# High-Dose-Rate Brachytherapy as an Organ-Sparing Treatment for Early Penile Cancer

**DOI:** 10.3390/cancers14246248

**Published:** 2022-12-19

**Authors:** Denisa Pohanková, Igor Sirák, Milan Vošmik, Linda Kašaová, Jakub Grepl, Petr Paluska, Lukáš Holub, Jiří Špaček, Miroslav Hodek, Martin Kopeček, Jiří Petera

**Affiliations:** 1Department of Oncology and Radiotherapy, Faculty of Medicine in Hradec Králové and University Hospital Hradec Králové, Charles University, Sokolská 581, 50005 Hradec Králové, Czech Republic; 2Department of Urology, Faculty of Medicine in Hradec Králové and University Hospital Hradec Králové, Charles University, Sokolská 581, 50005 Hradec Králové, Czech Republic; 3Department of Medical Biophysics, Faculty of Medicine in Hradec Králové, Charles University, Šimkova 870, 50003 Hradec Králové, Czech Republic

**Keywords:** penile carcinoma, HDR brachytherapy

## Abstract

**Simple Summary:**

Between 2002 and 2020, 31 patients with early penile cancer were treated with high-dose-rate brachytherapy (HDR-BT) at a dose of 18 × 3 Gy twice daily. The technique of the hollow steel needles fixed by the template was used. At a median follow-up of 117.5 months, the probability of local control at 5- and 10-years was 80.7% and 68.3%, respectively, with a probability of penis sparing of 80.6% and 62.1%. Cause-specific suervival was 100%. Only one case of radionecrosis was observed. These results show HDR-BT can achieve results that are comparable to low-dose rate BT while also allowing for organ preservation.

**Abstract:**

Background: Low-dose-rate brachytherapy is an effective organ-sparing treatment for patients with early-stage penile cancer. However, only limited data are available on the role of high-dose-rate brachytherapy (HDR-BT) in this clinical setting. Methods: Between 2002 and 2020, 31 patients with early penile cancer were treated at our center with interstitial HDR BT at a dose of 18 × 3 Gy twice daily. A breast brachytherapy template was used for the fixation of stainless hollow needles. Results: The median follow-up was 117.5 months (range, 5–210). Eight patients (25.8%) developed a recurrence; of these, seven were salvaged by partial amputation. Six patients died of internal comorbidities or a second cancer. The probability of local control at 5 and 10 years was 80.7% (95% CI: 63.7–97.7%) and 68.3% (95% CI: 44.0–92.6%), respectively. Cause-specific survival was 100%. Only one case of radiation-induced necrosis was observed. The probability of penile sparing at 5 and 10 years was 80.6% (95% CI: 63.45–97.7%) and 62.1% (95% CI: 34.8–89.4%), respectively. Conclusions: These results show that HDR-BT for penile cancer can achieve results comparable to LDR-BT with organ sparing. Despite the relatively large patient cohort—the second largest reported to date in this clinical setting—prospective data from larger samples are needed to confirm the role of HDR-BT in penile cancer.

## 1. Introduction

Penile cancer is a relatively rare malignancy, with an incidence in Europe and North America of less than one case per 100,000 [1]. In patients with early superficial lesions, numerous organ-preserving treatments are available, including gland resurfacing, laser therapy, tumor excision, and cryoablation, offering oncologic outcomes that are comparable to radical surgery while preserving sexual and urinary function, thus helping to maintain psychological health [2]. For more advanced tumors, the therapeutic approach depends on the tumor characteristics (size, histology, grade, and stage) and localization. Radical surgery (total or subtotal penectomy) is the most common treatment approach, with good local control rates and survival outcomes. However, the negative impact of surgical resection on sexual function can have a major detrimental psychological impact, leading to a significant deterioration in patients’ quality of life [3,4]. 

The primary alternative to surgery is radiotherapy, which offers the potential for organ preservation with tumor control rates that are comparable to radical surgery [5,6]. The main radiotherapy modalities are external beam radiotherapy (EBRT), interstitial brachytherapy, and surface mold brachytherapy [5]. EBRT achieves 5-year local control rates of 62% in T1-2 tumors and can spare patients from mutilating surgery in about 40% of cases [7]. Low-dose-rate brachytherapy (LDR-BT) has a long history as a treatment for early-stage penile cancer, with 5-year local control rates of 70–80% [1]. 

In the last two decades, high-dose-rate brachytherapy (HDR-BT) has largely replaced LDR-BT in the treatment of many cancers due to its better radiation safety profile, shorter treatment time (allowing for more patients to receive treatment in the same period of time), and a more precise dose distribution. Nevertheless, the radiobiological effects of HDR-BT differ from those of LDR-BT, with an increased risk of complications [8]. For this reason, it is critically important to select the optimal dose and fractionation schedule, which explains why HDR-BT has not been used for the treatment of penile cancer until relatively recently. To date, only six small studies [9,10,11,12,13,14] have evaluated clinical outcomes in patients treated with HDR-BT for penile carcinoma. Consequently, more data are needed to better characterize treatment outcomes and toxicity, as well as to define the optimal dose and fractionation schedule.

Our group previously reported preliminary clinical results in a small series of patients (*n* = 10) who underwent HDR-BT for penile carcinoma at our center [14]. The aim of the present study is to provide updated results from a larger patient cohort with a longer follow-up period. 

## 2. Materials and Methods

### 2.1. Patients

This prospective study (institutional ID FNHK-2001-01) included all patients (*n* = 31) who underwent HDR-BT for early-stage penile cancer at our center (Department of Oncology and Radiotherapy of University Hospital Hradec Králové, Czech Republic) from March 2002 to May 2022. The patients’ demographic and clinical characteristics are shown in Table 1. 

Tumor biopsies, circumcision, and ultrasound (or computed tomography [CT], as appropriate) of pelvic and inguinal nodes were performed in all patients. One patient with a stage cT1cN0M0 G2 tumor underwent adjuvant EBRT of the inguinal nodes. Another patient underwent inguinal lymphadenectomy at the time of biopsy and circumcision. In one patient with cT1cN1M0 G1, lymphadenectomy was performed after HDR-BT. 

Follow-up visits were performed every 3 months for 3 years, every 6 months for the next 2 years, and annually thereafter by the treating urologist and radiation oncologist. These visits consisted of a clinical examination with an ultrasound assessment of the pelvic and inguinal nodes. Patient-reported erectile function and sexual satisfaction were assessed verbally. Toxicity was graded according to the RTOG/EORTC (Radiation Therapy Oncology Group/European Organization for Research and Treatment of Cancer) radiation toxicity scoring system. 

All patients provided written informed consent with high-dose-rate brachytherapy.

### 2.2. Brachytherapy

#### Brachytherapy Technique

When we started using HDR-BT for penile cancer, no guidance or recommendations were available that we might follow, and we based our procedures on our experience with LDR-BT. The evaluation of the gross tumor volume (GTV) was based on inspection and palpation. After the GTV had been determined, a margin of 0.5–1 cm was added to create the clinical target volume (CTV). Brachytherapy was performed under general or spinal anesthesia. The first step was the insertion of a Foley catheter to assist in urethral localization. We used the technique of hollow steel needles (diameter 1.6 mm) to achieve a rigid geometry of the implant. Before the implantation, we planned the configuration of the needles, putting marks on the patients’ skin where the needles were to enter and exit to ensure coverage of the CTV according to the Paris system rules. The needles were inserted and fixed via an interstitial BT breast template with a square geometry and a separation of 10 mm between the needles. The needles were carefully positioned to avoid the urethra. The template consists of 2 square plates of plastics perforated by multiple holes separated at 1 cm intervals. The bridge keeps the 2 templates parallel at all times. A foam collar was placed around the penis to ensure the stable positioning of the organ during treatment (Figure 1). The number of planes (range, 1–3) was determined according to the individual characteristics of each patient. In most cases (*n* = 15), 2 planes with square patterns were used, followed by 3 planes (*n* = 9) and a single plane (*n* = 7). In 13 patients, an external plane of needles was inserted outside the penis through the template, with a bolus between the needles and the tissue to achieve optimal surface coverage. In 2 patients (the last 2 treated in this series), plastic tubes were used instead of needles. The mean duration of the entire procedure was 20 min. Intravenous corticosteroids were administered after the application to prevent postoperative edema.

Dose distribution in the first 2 patients was calculated with the Abacus GammaMed planning system (Gammamed, MDS Nordion, Haan Germany). In the other 29 patients, the BrachyVision planning system (Varian Medical Systems, Palo Alto, CA, USA) was used. Planning CT was precluded due to the artifacts caused by steel needles, and it was not possible to draw the GTV, CTV, and organs at risk (OAR). Because we started HDR-BT for penile cancer before any previously published experience with this method, we relied on the Paris school recommendation that in the case of a rigid application with needles and template, the imaging for dose calculation can be omitted. Such an approach assumes that a standard implant distribution has been achieved and maintained and that standard dose calculation can be used. Precise measurements accurate to within 1 mm must be taken of the spacing between the templates and of active source lengths. Dose calculations are then done for this stable cubic array. Consequently, we directly measured the distances between the two plates of the fixation template, between the needle tip and plate, between the needle tip and mucosa exit point, and between the needle tip and mucosa entry point. These data were used for the program for a square geometric pattern with a separation of 10 mm in the BrachyVision planning system. The reference points were specified 5 mm from the plane of needles and on the mucosa surface. Geometric optimization was used for the dose calculation. Hot spots were corrected by the dose shaper tool. A safety margin of ≧5 mm was used from the plane of needles to the urethra (Figure 2). The last 2 patients were treated with the plastic tubes technique with CT-based 3-dimensional (3D) planning. 

The prescribed dose was 18 fractions with 3 Gy per single fraction applied twice daily, with at least 6 hours interval between fractions. The dose distribution was optimized for reference points (5 mm from the plane of needles and on the mucosa surface). Ninety percent of the CTV should be covered by at least 90% of the prescribed dose (D_90_), and V_150_ should be less than 35% of the prescribed volume. The dose limits were prescribed for the urethra and the skin: the urethra < 100% of the prescribed dose, and the skin < 125% of the prescribed dose. The contour of the urethra was drawn into a dosimetric plan based on direct measurements of the distance of the urethra and the planes of needles. The treatment was delivered by HDR automatic afterloading device GammaMed (GammaMed, MDS Nordion, Haan, Germany). The step size of the iridium source dwelling position was 3 mm. 

The patients were admitted on Monday, and they underwent internal and anesthesiologic examinations. The brachytherapy application was performed the following day, and antiedema prevention was administered. On Wednesday, the isodose plan was calculated, and in the afternoon, the first fraction of HDR-BT was realized. The irradiation continued all working days. The day 16 of the hospitalization, the final (18th) fraction was delivered, the brachytherapy needles and urinary catheter were removed, and patients were discharged to home (Figure 3). 

### 2.3. Statistical Analysis

Basic descriptive statistics were used for the analysis, including median and/or means with standard deviation (SD) for continuous data and absolute and relative frequencies for categorical data. Survival data were analyzed using the Kaplan–Meier method with 95% confidence intervals (CI). The NCSS 8 statistical software program (NCSS, Keysville, UT, USA) was used for all statistical analyses.

## 3. Results

A total of 31 patients were included in this study (Table 1). The median age was 58.5 years (range, 33–72), and the median follow-up was 117.5 months (range, 5–210). The physical characteristics of brachytherapy applications are presented in Table 2.

Local recurrence was observed in eight patients (28.8%). The cumulative incidence of local recurrence was 31.7% (95%CI 7.4–56.0%). In two cases, the recurrence was out of the treated volume, on the opposite side of the glans. The median time to recurrence was 47 months (range, 7–98). Seven patients underwent partial amputation of the penis, with no further recurrences. One of these patients later developed nodal metastasis, which was treated with lymphadenectomy and EBRT; the patient is currently disease free. The eighth patient had both local and nodal recurrence, which was treated with partial amputation; the patient later died of a second tumor (lung carcinoma) with incomplete salvage for nodal disease. One patient developed regional recurrence to the inguinal lymph nodes without local failure. After lymphadenectomy and salvage EBRT, the patient is currently disease free. Of 31 patients, six died during follow-up, and 25 remain alive (80.1%). Of the six deceased patients, three died of another cancer (lung, head and neck, and brain, respectively), and three died of internal comorbidities. No other cancer was observed in the studied population. None of the patients died of penile carcinoma (Figure 4). 

Acute radiation toxicity was observed in all patients, most common grade (G)2 mucositis (100% of patients). The acute toxicity resolved in all cases within 8 weeks after completion of brachytherapy. All of the patients successfully completed the entire treatment without interruption. Late radiation toxicity included G1 angiectasia in seven patients (Figure 5) and urethral stenosis in two patients. One patient developed a postradiation ulceration requiring partial amputation of the penis. 

Local control (LC) rates at 5 and 10 years were 80.7% (95% CI: 63.7–97.7%) and 68.3% (95% CI: 44.0–92.6%), respectively (Figure 6). After salvage treatment, the LC rate was 100%. The disease-free interval (DFI) at 5 and 10 years was 73.0% (95% CI: 56.9–89.1%) and 63.2% (95% CI: 44.2–82.2%), respectively. The 5- and 10-year local recurrence-free interval (LRFI) was 82.1% (95% CI: 67.9–96.4%) and 72.3% (95 CI: 54.2–90.3%), respectively. Cause-specific survival (CSS) was 100%. 

Penile sparing was achieved in 22 patients; of these, two patients later died (one from a brain tumor and one from cardiac disease). Of these remaining 20 patients, most (17/20; 85%) continue to have an active sexual life that is comparable to their pretreatment status. The probability of penile sparing at 5 and 10 years was 80.6% (95% CI: 63.5–97.7) and 62.1% (95% CI: 34.8–89.4%), respectively (Figure 7). 

## 4. Discussion

In this study, we report treatment outcomes in a series of patients who underwent HDR-BT for penile cancer at our institution from 2002 to 2022. At a median follow-up of 117.5 months (range, 5–210), eight patients (25.8%) developed a recurrence requiring salvage surgery (partial amputation). The cause-specific survival rate was 100%, with six deaths due to other causes during follow-up. The probability of local control at 5 and 10 years was 80.7% and 68.3%, respectively, with penile sparing rates of 80.6% and 62.1%, respectively. Only one case of radiation-induced necrosis was observed. Overall, these results are comparable to those achieved with LDR-BT.

Iridium-192 LDR-BT has proven to be a successful organ-preserving treatment for early penile carcinoma, with numerous studies reporting LC rates ranging from 70%–80% [1]. Starting in the 1990s, HDR-BT began to replace LDR-BT for many cancer treatments, although its use for the treatment of penile cancer is more recent. In fact, to our knowledge, our group was the first to describe the application of HDR-BT for penile cancer in 2004 [15]. Since then, five other studies have been published (Table 3), although most of those studies were small, with a median of only 13 patients each (range, 7–76) and a relatively short follow-up (range, 20–90 months). Dose and fractionation schedules have also been highly variable, ranging from 3 to 5 Gy per fraction delivered over 5–9 days. 

The reported 5-year local recurrence-free survival rates for HDR-BT in penile cancer range from 65.6% to 86%, with 5-year CSS rates of 78%–88% (Table 3). In the two studies that used single doses ≥4 Gy, the incidence of necrosis was 9% and 10%, respectively [9,13]. By contrast, in the studies by Kellas-Slezcka et al. [10] and Sharma et al. [12], both of which used lower doses (3–3.5 Gy), the incidence of necrosis was 2.6% and 0%, respectively. Importantly, in the studies that used a higher dose per fraction [9,13], the risk of urethral stenosis was markedly higher, with Martz et al. [9] and Rouscoff et al. [13] reporting stenosis rates of 7% and 9%, respectively. By comparison, Kellas-Slezcka et al. and Sharma et al. –reported urethral stenosis rates of only 1.3% and 0%, respectively. Marbán et al. [11] reported extremely high rates of necrosis and urethral stenosis (43% each) in their series of seven patients treated with interstitial HDR-BT, despite administering single doses <3.75 Gy. According to those authors, the elevated treatment-related toxicity was due to significant dose inhomogeneity and higher skin doses. Overall, the reported results of the studies that have assessed HDR-BT for penile cancer appear to be comparable to those reported for LDR-BT [1]. 

In many countries, including the Czech Republic, surgical amputation remains the primary treatment for penile carcinoma, despite the availability of organ-preserving options such as brachytherapy. In fact, this tendency explains the relatively small number of patients treated with HDR-BT at our center over the study time period. Nevertheless, it is important to emphasize that the present study includes the second-largest cohort of patients with penile cancer treated with interstitial HDR-BT. Most of the patients in our series had an early disease: stage T1 or carcinoma in situ that recurred after previous surgical treatment. In the largest study published to date, Kellas-Sleczka et al. [10] evaluated treatment outcomes in 76 patients (stages T1, T2 and even T3), reporting 5- and 10-year LC rates of 65.6%, with organ preservation rates of 69.5% and 66.9%, respectively. The advantage of that study was the use of CT-based 3D planning in combination with the use of plastic tubes rather than steel needles, a technique that was first described in penile cancer by Sharma et al. [12]. The benefit of this approach is that it fully exploits the potential of HDR-BT to deliver a highly conformal dose distribution to the target. When our group first started using interstitial HDR-BT for penile cancer, we used rigid geometry with stainless steel needles fixed by template rather than the plastic tube technique. Only recently (our last two patients) have we started to use plastic tubes with CT-based 3D planning, which is the technique currently recommended by American and European guidelines [1]. To facilitate this technique, we created a special plastic bridge for tube fixation (Figure 8).

Acute toxicity in our series was mild and comparable with other studies (Table 3). The main challenge of penile brachytherapy, however, is late toxicity, particularly postradiation necrosis and urethral stenosis. In our study, two patients (6.5%) developed urethral stenosis, which was treated conservatively in both cases. We observed only one case (3%) of postradiation necrosis, probably due to the low dose homogeneity index (0.46). Kellas–Ślęczka et al. [10] reported an even lower incidence of necrosis (2.6%) and urethral stenosis (1.3%) in their large series. Those authors used single fractions <3.5 Gy, which was only slightly more than the 3 Gy used in our sample. Sharma et al. used [12] a similar fractionation scheme (42–51 Gy in 14–17 fractions BID), with no cases of necrosis or urethral stenosis. In line with the recommendations of the American Brachytherapy Society for interstitial HDR-BT of tongue cancer, we believe small fraction sizes are essential to reduce the risk of toxicity [24]. 

Despite the oncological efficacy of primary surgery for penile cancer, this approach often leads to major psychological problems due to organ mutilation. In fact, a substantial proportion of patients (≈40%) who undergo surgery experience a significant deterioration in quality of life, with 50% presenting long-term psychological problems and 60% developing reduced sexual function [25]. In this context, it is easy to understand the interest in alternative, organ-sparing approaches such as brachytherapy. The rationale supporting the use of brachytherapy for the treatment of early penile cancer is the potential for organ preservation (75% of cases), which allows patients to maintain quality of life without any substantial detrimental effect on survival. Moreover, in patients who undergo primary brachytherapy, salvage surgery (partial or total amputation) remains an option in case of local recurrence. Importantly, the available data show that this approach yields OS rates that are comparable to those achieved with primary surgery, as evidenced by the results of the meta-analysis by Hassan et al. [26], which included 2178 patients treated with either primary surgery (*n* = 1505) or primary brachytherapy (*n* = 676). Five-year OS rates in both groups were similar (76% after primary surgery and 73% after BT). Although penectomy achieved higher 5-year LC rates compared to brachytherapy (84% vs. 74%, respectively), this treatment approach involves greater morbidity. In patients with early-stage penile cancer treated with primary surgery, the 5-year OS and LC disease-free survival rates were 80% and 86%, respectively, versus 79% and 84% in patients treated with brachytherapy. 

### Strengths and Limitations

The main limitation of this study is the limited number of patients and the technique of hollow stainless needles. Contemporary standards of care are plastic tubes and 3D planning, ideally with MR guidance. By contrast, this is the second largest sample reported to date, with a relatively long follow-up (median, 117.5 months).

Compared to LDR-BT, HDR-BT is a relatively new technique for the treatment of penile cancer. Despite the limited clinical experience, the available data suggests that HDR-BT offers results that are comparable to (or even better) than conventional LDR-BT. Nevertheless, it is clear that more data are needed to refine the technique and to determine the optimal fractionation schemes. In this regard, the present study provides valuable data to expand our understanding of this promising technique. However, large, prospective—preferably multicentric—studies would be helpful to better determine the role of HDR-BT in the treatment of penile cancer. 

## 5. Conclusions

The results of this study suggest that HDR-BT for early-stage penile cancer provides results that are comparable to LDR-BT. Importantly, organ preservation was achieved in nearly three out of every four patients in our cohort. HDR-BT is a highly promising alternative to surgery and LDR-BT in well-selected patients. However, data from large, prospective studies are needed to better optimize this method. 

## Figures and Tables

**Figure 1 cancers-14-06248-f001:**
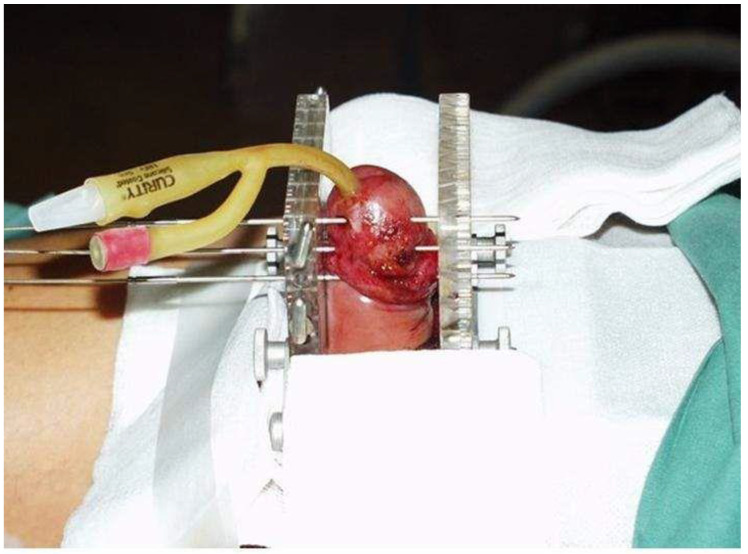
Implantation technique.

**Figure 2 cancers-14-06248-f002:**
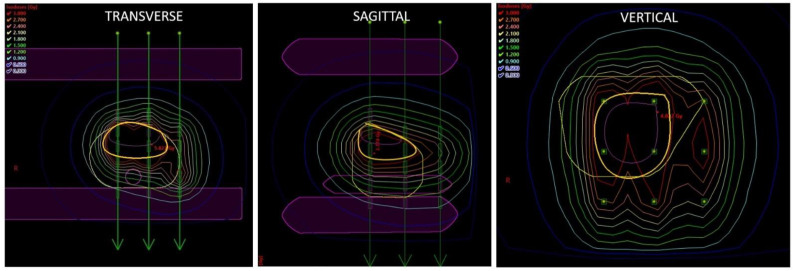
Dose distribution. The simple geometrical 3D model was created in the BrachyVision planning system (Varian Medical Systems, Palo Alto, CA, USA) by measuring the distance between both plates, between each needle tip and plate, and between plates and the musoca entry and exit point for each needle. The urethra was modeled as a straight tube. The width of the plates was 1 cm, and the needle separation was 10 mm in square grid geometry. Purple contour = GTV; ochre contour = CTV; yellow contour = penis surface; violet contour = urethra; violet-filled structures = plastic template.

**Figure 3 cancers-14-06248-f003:**
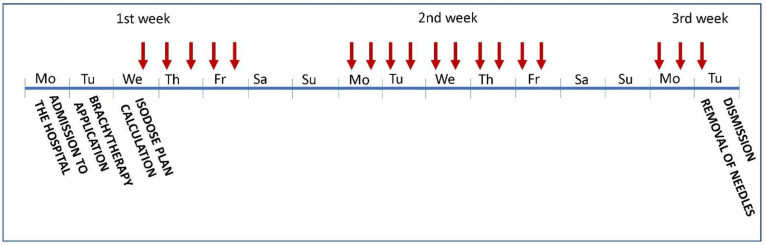
Scheme of treatment schedule. Red arrows indicate brachytherapy fractions.

**Figure 4 cancers-14-06248-f004:**
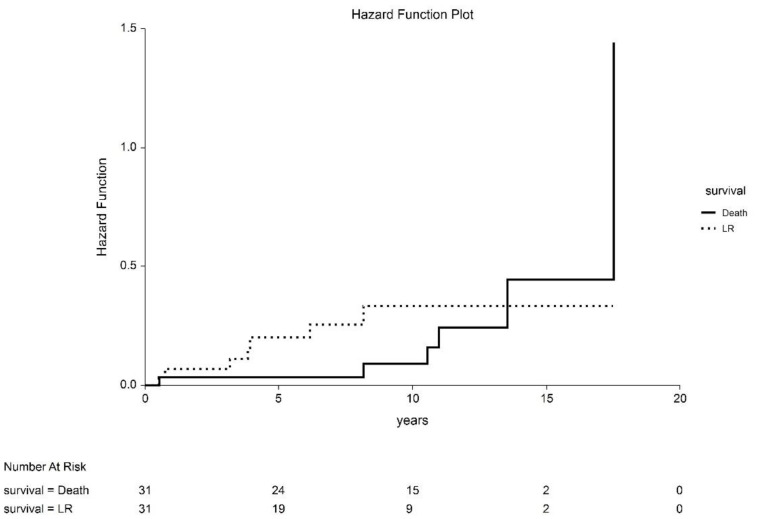
Competing risk analysis of deaths from other cancers.

**Figure 5 cancers-14-06248-f005:**
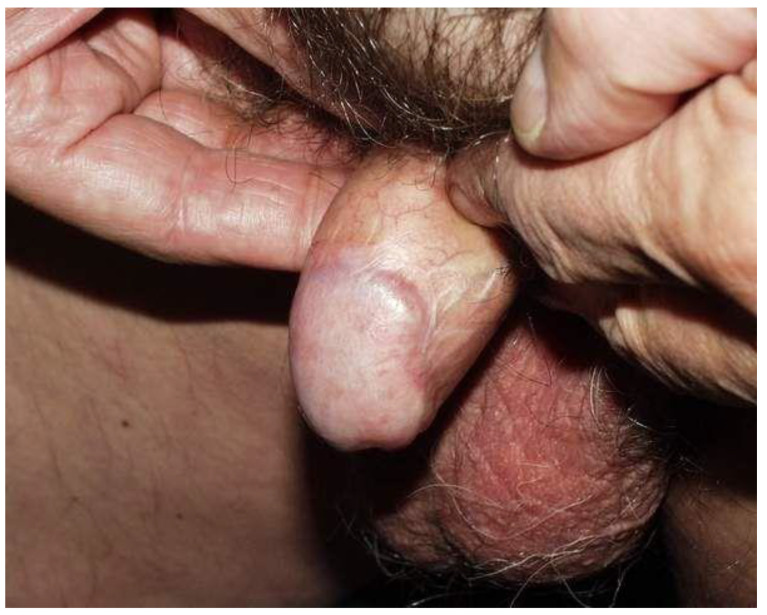
Result at 5 years.

**Figure 6 cancers-14-06248-f006:**
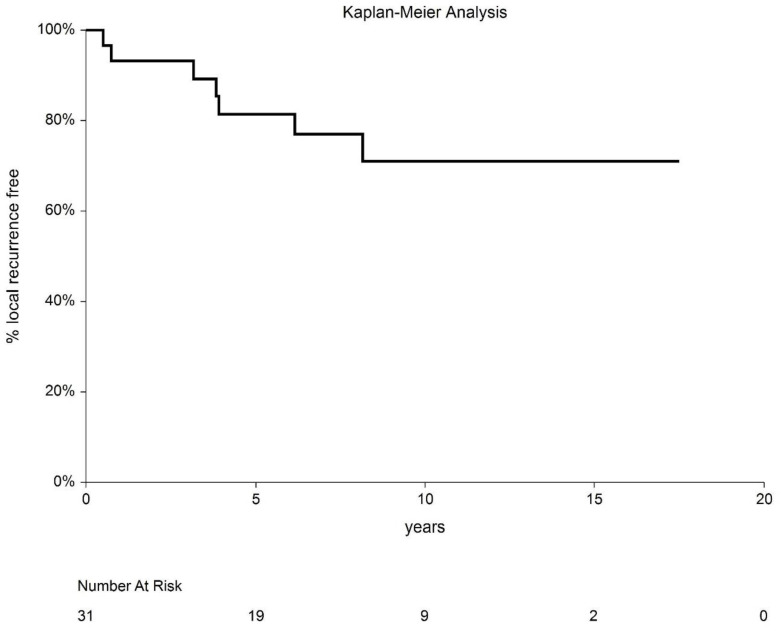
Local control recurrence-free interval after HDR-BT.

**Figure 7 cancers-14-06248-f007:**
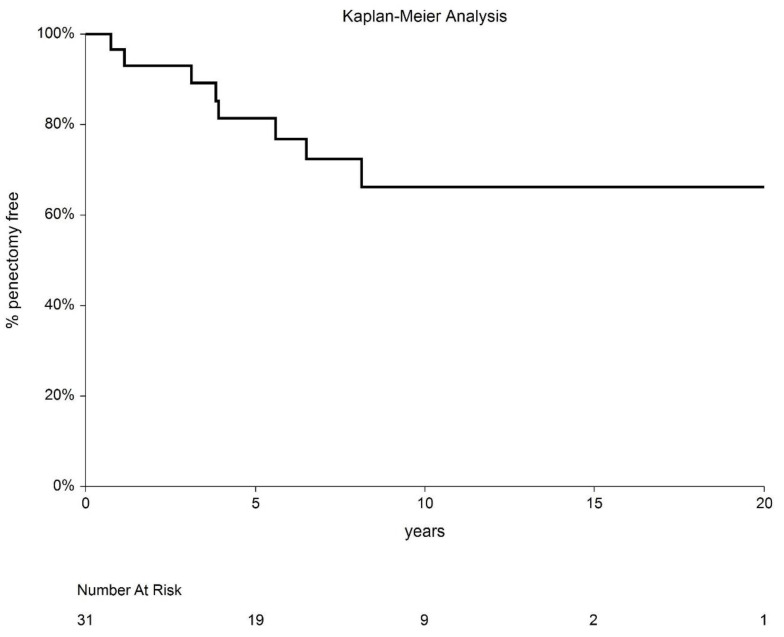
Penectomy-free interval outcomes.

**Figure 8 cancers-14-06248-f008:**
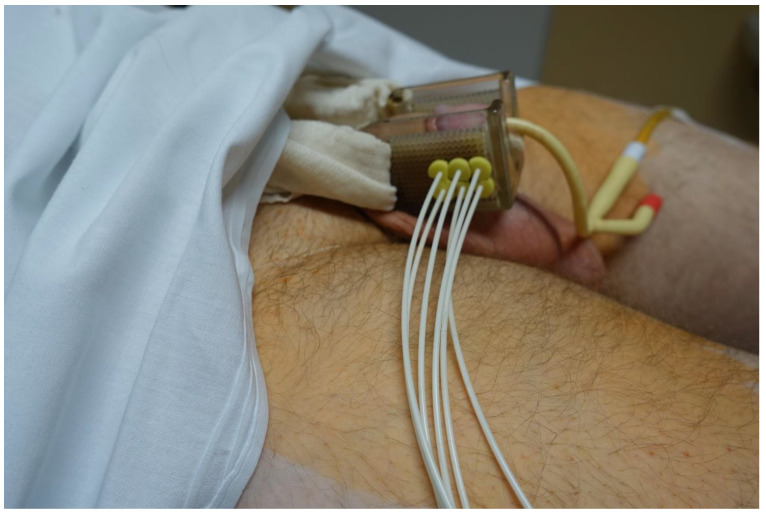
Template for CT-compatible brachytherapy.

**Table 1 cancers-14-06248-t001:** Patient characteristics.

Characteristic	*n*
Patients	31
Median age, years (range)	58.5 (33–72)
Tumor localization Glans Corpus	1 30
TNM TisN0M0 (recurrence after surgery) T1 N0M0 T1N1M0	6 24 1
Largest dimension, mm (range)	10.5 (5–30)
Median depth of invasion below the basement membrane at T1, mm (range)	2.0 (0.2–6)
Histology, squamous cell carcinoma	31
Tumor grade G1 G2 G3	22 8 1
Combined treatment Adjuvant irradiation of inguinal nodes Inguinal lymphadenectomy at the time of biopsy Inguinal lymphadenectomy after brachytherapy	1 1 1
Median follow-up, months (range)	117 (5–210)

**Table 2 cancers-14-06248-t002:** Physical characteristics of brachytherapy.

Characteristic	*n*
Number of catheters, median (range)	5 (2–12)
Number of planes, median (range)	2 (1–3)
Mean V100, cm^3^ (range)	6.56 (2.12–11)
Mean V150, cm^3^ (range)	2.26 (0.27–5)
Mean V200, cm^3^ (range)	1.14 (0–3)
Mean DHI (range)	0.64 (0.46–0.89)

Abbreviations: V100, volume irradiated by 100% of the prescribed dose; V150, volume irradiated by 150% of the prescribed dose; V200, volume irradiated by 200% of the prescribed dose; DHI, dose homogeneity index.

**Table 3 cancers-14-06248-t003:** Clinical results of main studies evaluating LDR-BT and HDR-BT for penile carcinoma.

Study	Patients, *n*	Dose, Gy	Mean FU, Months (Range)	5y-LRFS, %	5-y CSS, %	Complication Rates, Necrosis/Stenosis, %	% Penile Preservation/Years
**LDR-BT**							
Chaudhary et al. [16]	23	50	21 (4–117)	70	n.s.	0/9	70/8
de Crevoisier et al. [17]	144	65	68 (6–348)	80	92	26/ 29	72/10
Crook et al. [18]	67	60	48 (6–194)	87	83.6	12/ 9	88/5
Delannes et al. [19]	51	50-65	65 (12–144)	86	85	23/45	75/5
Kiltie et al. [20]	31	63.5	61.5	81	85.4	8/44	75/5
Mazeron et al. [21]	50	60–70	36–96+	78		6/19	74/5
Rozan et al. [22]	184	63	139	86	88	21/45	78/10
Soria et al. [23]	102	61–70	111	77	72	n.s.	72/6
**HDR-BT**							
Martz et al. [9]	29	35-39/9/F/5d BID	72.4	86	82	10.3/7	79.3/5
Kellas-Slezcka et al. [10]	76	42.8/48.2 3-3.5/F BID	76 (7–204)	65.6	85	2.6/1.3	66.9/10
Marbán et al. [11]	7	38.4/12 F—53.04/17F BID	90	86	86	43/43	86/7.5
Sharma et al. [12]	14	42-51/14-17 F BID	22	86	n.s.	0/0	93/3
Rouscoff et al. [13]	12	36/9 F/5d— 39/9 F/5d BID	27 (5.1–83)	83	100	9/9	92/5
Petera et al. [14]	10	54/18 F/9d BID	20 (3.4–90.6)	No recurrence during FU	100% during FU	0/0	100% during FU

Abbreviations: n.s., not specified; FU, follow-up; LDR, low-dose-rate; HDR, high-dose rate; LRFS, local recurrence-free survival; BID, twice daily; F, fraction; D, day; CSS, cancer-specific survival.

## Data Availability

The data presented in this study are available on request from the corresponding author.

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
