# Peer review of "High-Dose-Rate Brachytherapy as an Organ-Sparing Treatment for Early Penile Cancer"

_cancers, 2022, doi:10.3390/cancers14246248_

Round 1
Reviewer 1 Report
The authors review a large (31 patients) cohort of patients treated with HDR-BT for penile cancer. This is a large cohort for a rare malignancy that is typically treated surgically. Their control rates are similar to older cohorts treated with LDR-BT or PDR-BT. The manuscript is well purposed. Worded well and reports findings objectively. I believe it is appropriate for this journal given the potential for educating a wider audience.
Introduction:
No specific comments
Materials and methods:
Why did the authors not treat patients on a prospective outcomes study? This is a relatively novel treatment and doing so should be justified in the introduction if you are not treating on a study protocol. Furthermore, what was the "written informed consent" for?
Planning: The technique is outdated. In the limitations section the authors should acknowledge that 3D planning ideally with MR guidance is standard of care.
Results:
It is difficult to interpret what the clinical target volume would be based on the description given. Were the local failures marginal misses?
% local recurrence is not an appropriate outcome measure. They should present cumulative incidence of local recurrence instead. Given the deaths from other cancers is highly present in this population then a competing risk analysis is warranted.
Discussion: The authors present an appropriate discussion of prior studies exploring brachytherapy treatment for penile cancer.
Reviewer 2 Report
In this study, 31 patients with early penile cancer were treated with high-dose-rate brachytherapy (HDR-BT) at a dose of 18 x 3 Gy twice daily and followed up to relative a long period of time (median 117.5 months, range, 5-210).
Comments:
The manuscript was well written and makes sense, major revision was needed.
Major revision
1. Delineation of gross target volume and dose distribution should be shown in a typical figure, if possible.
2. Combined or previous treatment should be explained and listed in the table. For patients with node positive, whether lymph node dissection or chemotherapy was used? As only one patient with cT1cN1M0 G1, lymphadenectomy was performed after HDR-BT.
3. Definition section was needed, eg. for HDR, LDR and acute radiation toxicity, and others.
Minor revision
1. The description “interstitial HDR BT at a dose of 18 x 3 Gy twice daily” was not clear, which is Gy/fraction? Please specify.
2. Please recheck grammar and punctuation, eg. the patient is currently disease free
Reviewer 3 Report
To my opinion, the authors adequately answered all the questions raised by the reviewers.
